# Interleukin-6: Cardiovascular Aspects of Long-Term Cytokine Suppression in Patients with Rheumatoid Arthritis

**DOI:** 10.3390/ijms252212425

**Published:** 2024-11-19

**Authors:** Elena V. Gerasimova, Tatiana V. Popkova, Irina G. Kirillova, Daria A. Gerasimova, Evgenii L. Nasonov, Aleksandr M. Lila

**Affiliations:** 1Department of Systemic Rheumatic Diseases, V.A. Nasonova Research Institute of Rheumatology, Kashirskoe Shosse, 115522 Moscow, Russia; popkovatv@mail.ru (T.V.P.); dr.i.kirillova@yandex.ru (I.G.K.); nasonov@irramn.ru (E.L.N.); amlila@mail.ru (A.M.L.); 2Chair of Organization and Economy of Pharmacy, Institute of Pharmacy, A.P. Nelyubina, I.M. Sechenov First Moscow State Medical University (Sechenov University), 96k1 Ave. Vernadsky, 119526 Moscow, Russia; gerasimova_d_a@staff.sechenov.ru; 3Russian Medical Academy of Continuous Professional Education of the Ministry of Healthcare of the Russian Federation, Build 1, 2/1 Barrikadnaya St., 125993 Moscow, Russia

**Keywords:** interleukin-6, tocilizumab, traditional risk factors, dyslipidemia, mSCORE, cIMT, sVCAM

## Abstract

In recent years, many atherogenesis researchers have focused on the role of inflammatory cytokines in the development of cardiovascular disease (CVD). Interleukin-6 (IL-6) cytokine is independently associated with higher CVD risk in patients with rheumatoid arthritis (RA). The effect of IL-6 inhibitors on the cardiovascular system in RA patients remains poorly understood, especially with its long-term use. This study investigates the effect of therapy with IL-6 receptor blocker tocilizumab (TCZ) on the dynamics of cardiovascular risk (CVR), modifiable risk factors (RFs), carotid artery (CA) structural changes, and the incidence of cardiovascular complications (CVCs) in RA patients during a 265-week follow-up period. Forty-five patients with active RA (DAS28-ESR 6.2 (5.5;6.8) with ineffectiveness and/or intolerance to disease-modifying antirheumatic drugs (DMARDs) were included in this study. During long-term therapy with TCZ in RA patients, no increase in CVR and no significant structural changes in CA were observed. No significant changes in the blood lipid spectrum were observed in patients without statin therapy. In the group of patients receiving statins, there was a 43% increase in high-density lipoprotein cholesterol (HDL-C), a 15% reduction in total cholesterol levels, and a 56% decrease in the atherogenicity index (*p* < 0.01 in all cases). Associations were found between ∆ total cholesterol and ∆ C-reactive protein (CRP) (R = 0.36, *p* = 0.04), ∆ low-density lipoprotein cholesterol (LDL-C), and ∆-CRP (R = 0.42, *p* = 0.03) in RA patients receiving statins. Initially, the thickness of the intima–media complex of carotid arteries (cIMT) positively moderately correlated with age (R = 0.7; *p* < 0.01), BMI (R = 0.37; *p* < 0.01), and systolic blood pressure (R = 0.64; *p* < 0.01); however, it weakly correlated with the lipid spectrum parameters: total cholesterol (R = 0.29; *p* < 0.01) and LDL-C (R = 0.33; *p* < 0.01). No new associations of cIMT by the end of the follow-up period, as well as the relationship of cIMT value with RA activity and therapy, were revealed. Patients with carotid ASPs showed an oppositely directed relationship between total cholesterol and sVCAM-1 at baseline (R = −0.25, *p* = 0.01) and at the end of this study (R = 0.29, *p* < 0.01). The incidence of cardiovascular events was 0.53 per 100 patient-years during the 265-week period of TCZ therapy.

## 1. Introduction

Rheumatoid arthritis (RA) is a disease with high cardiovascular risk (CVR). Despite advances in diagnostics and therapy, mortality from cardiovascular disease (CVD) in RA remains high [1]. It has been established that premature mortality in RA is associated with the development of cardiovascular complications (CVCs) due to the progression of atherosclerosis [2,3].

The causes of accelerated development of atherosclerosis-associated CVD in RA patients include accumulation of traditional cardiovascular risk factors (RFs), chronic inflammation, and side effects of antirheumatic therapy [4,5]. Interleukin (IL)-6 is the leading pro-inflammatory cytokine, the hyperproduction of which can influence blood lipid levels and the progression of atherosclerotic vascular lesions [6,7]. Polymorphisms in the gene encoding IL-6R and reducing IL-6 activity were associated with a low risk of CVD and a favorable cardiometabolic profile [8,9,10]. According to the post-analysis CANTOS (Canakinumab Anti-Inflammatory Thrombosis Outcomes Study), canakinumab has a cardioprotective effect in patients with a significant reduction in IL-6 levels, suggesting that inhibiting IL-6 signaling may reduce CVR [11].

The imbalance of cytokines and the accumulation of inflammatory mediators contribute to the development of vascular disorders associated with atherogenesis: endothelial dysfunction, vasoconstriction, lipid and lipoprotein peroxidation, hypercoagulation, and later lead to the formation and destabilization of atherosclerotic plaques (ASPs) and the development of CVCs [12,13]. It was shown that in the assessment of early atherosclerotic vascular changes, the prognostic significance of adhesion molecules—soluble intercellular adhesion molecule-1 (sICAM-1) and soluble vascular cell adhesion molecule-1 (sVCAM-1)—was significantly higher compared to other biological mediators [14,15]. 

Early initiation of effective anti-inflammatory therapy is crucial in the prevention of CVD, which not only modifies the course of the disease but also reduces the risk of cardiovascular catastrophes [16,17].

In this context, the study of the cardiovascular effects of biologics is of undoubted interest. On the one hand, they have an “anti-atherogenic” effect by suppressing the inflammatory component of atherothrombosis. On the other hand, they may affect the vascular wall and cholesterol transport in blood [18,19], thereby affecting the risk of cardiovascular events. Tocilizumab (TCZ) (a humanized monoclonal antibody blocking the IL-6 receptor) is a promising drug successfully used for the treatment of patients with high RA activity. The long-term efficacy and safety of TCZ therapy in RA patients have been demonstrated in a number of studies [20,21]. A known side effect of TCZ is an increase in total cholesterol, low-density lipoproteins cholesterol (LDL-C), and high-density lipoprotein cholesterol (HDL-C) levels with short-term use of the drug, with less data with long-term use [22,23]. The influence of IL-6 inhibitors on the cardiovascular system in RA patients remains underinvestigated, especially with its long-term use. This research aims to investigate the effect of TCZ therapy on the dynamics of modifiable RFs, CVR, lipids profile, structural changes in the carotid arteries (CAs), and frequency of CVC in RA patients during a 265-week follow-up period.

## 2. Results

### General Characteristics of the Study Population

Thirty-seven RA patients were included in this study. The majority (86%) were female, with a median age of 56 (48;68) years and an RA duration of 92 (49;158) months. RA activity was high (DAS28-ESR > 5.6, SDAI > 26 or CDAI > 22) in all patients, and median DAS28 was 6.2 (5.7;6.7), SDAI was 35 (29;41), and CDAI was 31 (24;37). More than half (59%) of the patients had extra-articular manifestations. All patients were seropositive for IgM rheumatoid factor (RF) and 86% of patients were seropositive for antibodies to cyclic citrullinated peptide (anti-CCP) antibodies.

All patients had a history of treatment failure or intolerance to two or more disease-modifying antirheumatic drugs (DMARDs). TCZ was administered in a dose of 8 mg/kg intravenously with an interval of 4 weeks, with 51% receiving monotherapy and 49% receiving a combination with DMARDs. In some patients (60%), intravenous administration of the drug was later changed to subcutaneous administration at a dose of 162 mg once a week. The median duration of the course of intravenous infusions in these patients was 192 (176;210) weeks. At the time of inclusion in this study, 19 patients (51%) were taking glucocorticoids (median prednisolone dose was 7.5 (5.5;10.4) mg/day), 25 patients (68%) were taking nonsteroidal anti-inflammatory drugs (NSAIDs), and seven patients (19%) were taking statins. An additional 10 patients were prescribed atorvastatin at a dose of 10 mg/day after inclusion in this study. The median follow-up period was 265.4 (251.5;283.4) weeks. The general characterization of patients is presented in Table 1.

By the end of this study, 87 percent of RA patients achieved remission (DAS28 < 2.6, SDAI ≤ 3.3, or CDAI ≤ 2.8). Low disease activity (2.6 < DAS28 < 3.1, SDAI 3.3-11, or CDAI 2.8–10) was found in 13% of cases after 265 weeks of TCZ therapy. By the end of this study, there was a significant decrease in DAS28, HAQ, C-reactive protein (CRP), erythrocyte sedimentation rate (ESR), sVCAM-1, and serum IL-6 concentration (Table 2).

At week 265 of this study, no patient was receiving nonsteroidal anti-inflammatory drugs (NSAIDs) and 13 of 23 patients (57%) were able to discontinue glucocorticoids (*p* < 0.01).

At the time of inclusion in this study, dyslipidemia was registered in 30 patients (67%), arterial hypertension in 29 (64%), overweight in 23 (51%), family history of CVD in 16 (36%), and smoking in 7 (16%). Every third patient was found to have a combination of three or more traditional RFs. Among the manifestations of dyslipidemia, hypercholesterolemia was detected in 67% of RA patients, increased LDL-C level in 67%, decreased HDL-C level in 33%, and hypertriglyceridemia in 20%. Three cases (7%) of type 2 diabetes mellitus (DM) were reported among the RA patients. There were also two cases (7%) of class I coronary artery disease (CAD) (one of these patients had a non-ST-segment elevation myocardial infarction (MI) more than 5 years ago and one coronary stenting more than 2 years ago) and two cases (7%) of chronic heart failure (CHF) class I-II NYHA. After 265 weeks, the modifiable RFs (dyslipidemia, arterial hypertension, overweight, and smoking) occurred with the same frequency (Table 3).

There was a decrease in the number of patients with hypercholesterolemia and with reduced blood HDL-C content. The number of patients with elevated LDL-C and triglyceride (TG) levels did not change reliably during the follow-up period. By the end of this study, the majority (62%) of the patients showed an increase in body mass index (BMI), five of them were no longer underweight, and six of them became overweight. Four patients stopped smoking, but this had no significant effect on the prevalence of this factor at re-examination.

Two groups were formed to analyze the dynamics of the blood lipid spectrum: patients of the first group received statins (*n* = 21), while in the second group, such treatment was not performed (*n* = 24). At the end of the follow-up period in group 1, there was a 15% decrease in total cholesterol level and a 43% increase in HDL-C; LDL-C and TG concentrations did not change (Table 4).

We did not document any adverse events (AEs) associated with statin therapy in RA patients receiving 265 weeks of TCZ. No significant fluctuations in the blood lipid spectrum were observed in group 2. Almost one-third (35%) of patients in group 2 had negative dynamics of blood lipid spectrum (increase in LDL-C and TG concentrations by 36–46%; decrease in HDL-C level by 15–23%). Such dynamics are mostly leveled by the group due to some improvement (by 10–15%) in the lipid spectrum in some patients. In group 1, a correlation was found between ∆CRP and ∆ total cholesterol (R = 0.36, *p* = 0.04) and ∆CRP and ∆LDL-C (R = 0.42, *p* = 0.03), while in group 2, similar relationships were not found.

A high prevalence of carotid ASPs was found among RA patients initially. In 24 cases (53%), one atherosclerotic plaque (ASP) was detected, and in six cases (13%), two ASPs were found. During this study, three (7%) patients were found to have new ASPs: two for the first time and one had a new ASP in the right CA (Table 5).

No reliable differences in cIMT and frequency of carotid ASP detection in patients of groups 1 and 2 were observed either at the time of inclusion or at the end of observation.

Initially, cIMT was positively moderately correlated with age (R = 0.7; *p* < 0.01), BMI (R = 0.37; *p* < 0.01), and systolic blood pressure (SBP) (R = 0.64; *p* < 0.01); however, it was weakly correlated with the lipid spectrum parameters: total cholesterol (R = 0.29; *p* < 0.01) and LDL-C (R = 0.33; *p* < 0.01). There were no new associations of cIMT by the end of the follow-up period, as well as no new associations of cIMT with RA activity indicators (DAS28, CRP, and ESR) or conducting therapy. Correlation analysis did not reveal the association between the cIMT and blood concentrations of soluble adhesion molecule (sICAM-1 andsVCAM-1) and IL-6 in RA patients before therapy and after 265 weeks of this study. Also, the levels of sICAM-1, sVCAM-1, and IL-6 were comparable in RA patients with and without carotid artery atherosclerosis, both at the beginning and at the end of the observation. Patients with carotid atherosclerosis showed an oppositely directed relationship between total cholesterol and sVCAM-1 at baseline (R = −0.25, *p* = 0.01) (a) and at the end of this study (R = 0.29, *p* < 0.01) (b) (Figure 1).

By the end of this study, there was no significant change in the distribution of patients according to the Systematic Coronary Risk Evaluation (mSCORE) value. Median mSCORE was 2.3% at the beginning of this study and 3.3% at the end, *p* > 0.05. Very high CVR on the mSCORE scale was found in 33 (73%) patients at baseline, and in 28 (62%) patients it was due to diagnosed atherosclerosis of CA and/or type 2 DM with target organ damage. By the end of this study, this risk group included one additional person (mSCORE increased from 7% to 11%). Single patients were found to have moderate (5% and 8%) and low (17% and 11%) CVR at the beginning and end of this study, respectively.

The development of MI was registered at the 22nd week of TCZ therapy in a 66-year-old man with class II angina pectoris, which required emergency X-ray endovascular treatment (percutaneous balloon angioplasty and stenting of CA). Thereafter, the patient was continued on combination therapy with TCZ and methotrexate. Another 68-year-old female patient developed unstable angina pectoris at 24 weeks of TCZ therapy, which was resolved by conservative treatment. The other patients had no clinical symptomatology of CAD during a 260-week follow-up period. No cases of stroke were documented in the patients. Thus, the incidence of CVCs was 0.53 per 100 patient-years over the 265-week period of TCZ therapy.

## 3. Discussion

Previous [24] and current research results have shown the high efficacy of TCZ therapy in RA patients. Remission according to DAS28-ESR was observed in 64% of patients after 12 months and in 86% of patients after 5 years of TCZ therapy. The results of the 5-year AMBITION trial confirm the sustained and increasing efficacy of TCZ monotherapy over time: the proportion of patients achieving clinical remission by DAS28 after 24 and 264 weeks was 40.2% and 65.2%, respectively [19].

Researchers have proven the safety of short-term use of IL-6 inhibitors in relation to the cardiovascular system [18,19,25]. There is much less data on the effect of long-term therapy.

In our study of RA patients, traditional RFs before and after 265 weeks of TCZ therapy were encountered with the same frequency. Also, CVR to the mSCORE did not change during the follow-up period. By the end of this study, RA patients had a significant increase in BMI (11% on average). This result correlates with the data of other studies on the dynamics of anthropometric indices against the background of therapy with IL-6 inhibitors [26,27]. Choi I. et al. [26] observed an increase in body weight by 0.7 kg, 1.2 kg, and 1.1 kg in RA patients after 24, 48, and 72 weeks of TCZ treatment, with an average percentage weight change from baseline of 1.3%, 2.2%, and 2.0%, respectively.

Weight gain in RA patients on TCZ therapy has been shown to be due to an increase in muscle mass but not fat mass [27,28]. In the ADIPRAT study [29] using dual-energy X-ray absorptiometry in 107 patients with RA, a significant increase in total fat-free mass of 1 kg and 1.3 kg was found after 6 and 12 months of TCZ therapy, respectively (*p* < 0.05).

IL-6 is known to be a myokine, i.e., a cytokine produced and released by myocytes under the action of contractile activity. Inhibition of IL-6 may lead to a reduction in muscle atrophy [30]. Elevated IL-6 levels are known to be associated with cachexia, and IL-6 inhibitors are used as adjuvants in the treatment of cancer patients to prevent this catabolic effect [31]. Application of an anti-IL-6 antibody to an experimental tumor-bearing mouse model suppressed IL-6 secretion in the brain, which was accompanied by an attenuation of cachexia and hyperactivity in the area postrema network and prolonged the life of the mice [32].

The majority of studies [18,33,34,35,36] and meta-analyses [19,37,38] indicate an increase in blood lipid levels during therapy with IL-6 inhibitors in patients with inflammatory joint diseases.

A pooled report on the safety of TCZ based on five randomized controlled trials (RCTs) and their long-term extension studies (AMBITION, RADIATE, TOWARD, OPTION, and LITHE) demonstrated that an increase in serum lipids in patients was observed as early as 6 weeks after the first TCZ infusion and remained at this level for 104 weeks of treatment [36].

Later, G. Navarro et al. [37], summarizing the data of RCTs (AMBITION, TOWARD, and LITHE), noted that 33% of patients with RA receiving TCZ at a dose of 8 mg/kg had a clinically significant increase in total cholesterol and LDL-C levels. In our RA patients not receiving statins, after 260 weeks of TCZ use, no adverse changes in blood lipid spectrum were recorded, but about one third of them had a significant increase in LDL-C and a decrease in HDL-C. In M. Soubrier et al.’s study [34], 30% of patients not receiving statins after 2 years of TCZ therapy showed an increase in LDL-C levels from <3.4 mmol/L at the beginning of this study to ≥3.4 mmol/L by the end of this study.

At present, it remains unclear whether elevated lipid levels affect CVR in RA patients receiving TCZ. However, the benefit of hypolipidemic therapy in these patients is obvious and beyond doubt. This is clearly demonstrated when analyzing the dynamics of blood lipids in RA patients depending on the therapy with statins. In the TOWARD study [33], 16 patients with RA receiving TCZ and statins showed a 1.9 mmol/L decrease in total cholesterol concentration after 24 weeks of observation. The authors note that concomitant administration of TCZs and statins is most effective in reducing LDL-C levels.

A retrospective analysis of the effects of statins in RA patients treated with TCZ for 2 years also showed a stabilizing effect of statins on lipid levels without a clinically significant increase in AEs [34]. Patients who were simultaneously treated with TCZ and statins showed a persistent decrease in LDL-C levels, while patients who took statins before starting TCZ therapy or never received them showed a persistent increase in LDL-C levels after 3–4 months of treatment. Similar trends were observed for total cholesterol, HDL-C, and TG.

It can be assumed that timely administration of statins to RA patients with dyslipidemia improves the blood lipid spectrum. Thus, the number of patients with hypercholesterolemia and reduced HDL-C levels decreased more than three times by the end of our study. AI decreased almost two times. The positive associations of the dynamics of total cholesterol, LDL-C, and CRP levels in the group of patients receiving statins may indicate the anti-inflammatory effect of statins.

A systematic review and meta-analysis of 15 RCTs [39] demonstrated a reduction in CRP, ESR levels, and disease activity according to DAS28 in RA patients receiving statins compared to patients without hypolipidemic therapy. There is evidence of a decrease in the concentration of IL-6 and tumor necrosis factor (TNF)-α during statin therapy in RA.

A Mendelian randomized analysis showed that the anti-inflammatory effect of statins on coronary heart disease does not involve the IL-6 signaling pathway [40]. Presumably, the anti-inflammatory effect of statins is realized through the reduction in endothelial oxidation products, inhibition of leukocyte and endothelial cell adhesion, reduction in pro-inflammatory cytokines IL-6, IL-1β, and TNF-α, and ASP stabilization [41,42,43]. The ability of statins to modulate the expression of tissue factor, thrombin generation, and most thrombin-catalyzed procoagulant reactions explains their anticoagulant effect [44].

There is evidence of a greater CVR reduction with statin therapy in individuals with higher levels of circulating markers of inflammation, such as CRP levels [45].

On the other hand, the correlation of CRP concentration with total cholesterol and LDL-C levels found in our work suggests that disease activity may have a direct effect on blood lipid profile in RA patients. A similar relationship between changes in disease activity (DAS28 and CDAI) and blood lipid levels shown in other studies confirms the metabolic effects of systemic inflammatory disorders [23,46].

In addition, TCZ therapy can improve the qualitative and functional lipid parameters (including favorable modification of very low-density lipoprotein (VLDL) particles, recovery of the anti-atherogenic function of HDL-C, and reduction in LDL-C, reduce the amount of serum amyloid A and secretory phospholipase A2-IIa, and increase serum paraoxonase-1 [47]. IL-6 receptor blockade affects the enzyme proprotein convertase subtilisin/kexin type 9 (PCSK9), which plays an important role in total cholesterol homeostasis [48]. The favorable effect of TCZ on lipid clearance in the liver can be evidenced by the increased expression of the LDL receptor on liver cells cultured with IL-6 detected in in vitro experiments [49].

In addition to changes in blood serum lipids and lipoproteins, several studies [50,51,52] demonstrate improvement in endothelial function and prothrombotic status, reduction in pulse wave velocity with improvement in arterial stiffness, improvement in oxidative stress, suppression of monocyte inflammatory profile, and nethosis formation in patients receiving TCZ.

The influence of cytokines extends not only to the joints but also to the blood vessels, and the involvement of coronary arteries accounts for a significant proportion of cardiovascular events in active RA [53]. Prolonged synovial secretion of pro-inflammatory cytokines leads to chronic vascular endothelial activation and macrophage dysfunction [54]. The creation of a proatherogenic environment promotes atherosclerotic disease due to endothelial dysfunction and autoantibodies in RA that increase the inflammatory potential of macrophages [55]. Activation of adaptive immunity culminates with prolonged synthesis of pro-inflammatory cytokines and continuous recruitment of inflammatory cells into tissues.

Although the role of VCAM-1 in the cardiovascular system is not fully understood, several studies have shown that VCAM-1 is expressed in various CVDs such as atherosclerosis, CAD, stroke, MI, CHF, and arterial hypertension [15,56,57]. Determination of sVCAM-1 added prognostic value to classical risk factors and CRP in relation to CVD risk in the general population [58,59]. Evidence suggests that sVCAM-1 may reflect ongoing proatherogenic processes in RA patients. A positive association between subclinical atherosclerosis progression and sVCAM-1 level in early-stage RA has been demonstrated [60]. Apparently, sVCAM-1 plays a less important role in progressive atherosclerosis than in early atherogenesis. In our and other studies [61], no significant difference in sVCAM-1 levels was found between patients with and without carotid ASPs.

In Colunga-Pedraza I. et al.’s study [61], higher sVCAM-1 titers were associated with lower total cholesterol, LDL-C, and HDL-C in RA patients with and without carotid ASPs. A similar negative correlation of total cholesterol and sVCAM in RA patients with carotid atherosclerosis in our study was found at baseline. It is significant that at the end of this study, in the absence of active inflammation in our patients, a positive relationship was observed between total cholesterol and sVCAM-1. Whereas the correlation between sVCAM-1 and total cholesterol was positive in people without rheumatic diseases [62]. Thus, the paradoxical relationship between blood lipids and inflammation in RA, termed the "lipid paradox", may depend on a high sVCAM-1 level. Information has been obtained on the anti-atherosclerotic effect of a number of anti-inflammatory drugs associated with the inhibition of the expression of sVCAM-1 and sICAM-1 [63,64].

In the studies on cIMT dynamics in TCZ patients, no significant changes were recorded [65,66]. The results of our work showed the absence of negative dynamics of CA state in RA patients under TCZ treatment for 5 years. In a few patients (8%), new carotid ASPs were detected. In contrast, in our earlier study [67] in which we evaluated the effect of 24-week TCZ therapy, new ASPs were detected at a higher frequency (in 24% of RA patients). Such negative dynamics could be due to high RA activity in the first months of observation and the short duration of TCZ therapy since it is the disease activity that leads to the development of cardiovascular system damage [68]. This is also evidenced by the two cardiovascular events that developed in our patients during the first 6 months of TCZ therapy and no CVCs during further use of the drug.

To conclude that it is the long-term use of IL-6 inhibitors on the progression of atherosclerotic process that has a favorable effect on the progression of atherosclerotic process allows the association of long-term average (measured over 3 years), rather than baseline IL-6 concentration, with the increase in cIMT [69].

In population studies, a positive correlation of IL-6 levels with the presence of unstable ASPs, the dynamics of their morphologic changes, and the degree of carotid stenosis was shown [7,70]. In patients with early-onset CAD, IL-6 levels were negatively correlated with ascending aortic diameter, left ventricular ejection fraction, and right ventricular end-diastolic diameter [71]. These changes may contribute to right ventricular remodeling and left ventricular systolic dysfunction. Blockade of IL-6 signaling using spg130Fc, along with suppression of synovitis, improved vascular function in mice with collagen-induced arthritis [60].

A systematic review [19] and several studies [72,73] have demonstrated encouraging data on the risk of serious adverse cardiovascular events with TCZ therapy compared to other biologics. In J. Zhang et al.’s study [72], the risk of MI in RA patients treated with TCZ was 36% lower than with abatacept. In another study, TCZ treatment resulted in fewer cardiovascular events than TNF-α inhibitors (hazard ratios, HR 0.68; 95% confidence interval, CI 0.49 to 0.94) among RA patients who switched from a different biologic drug or tofacitinib to TCZ versus a TNF-α [47]. Castagné B. et al. [19] noted that the use of TCZ demonstrates a favorable cardiovascular profile despite a more pronounced increase in total cholesterol and LDL-C levels than with other biologics.

Cardiovascular events and venous thromboembolism (VTE) were assessed in a recently published systematic review including six observational studies and one RCT [74]. The risk of major adverse cardiac events (MACEs), MI, stroke, CHF, and coronary revascularization in RA patients treated with TCZ did not exceed that in patients treated with other biologics (TNF-α inhibitors, abatacept, and rituximab). In the ENTRACTE randomized controlled trial evaluating the safety of TCZ and etanercept (ETN) for RA patients, 83 MACEs occurred during the follow-up period in patients receiving TCZ compared with 78 in the ETN group. The calculated HR of MACE occurrence in the TCZ group compared with the ETN group was 1.05 (95% CI 0.77–1.43) [75]. The incidence of VTE was sporadic in both groups and no increased risk was found in patients treated with TCZ compared to patients treated with ETN.

A large prospective study of the efficacy and safety of 2-year TCZ therapy in elderly patients found an increased incidence of severe cardiovascular events in the older age group of RA patients [76]. Thus, MI or sudden coronary death was observed in single (0.3–0.7%) patients ≤65 years of age and in 1.2% of patients >65 years of age, *p* = 0.011. It is possible that the higher incidence of AEs in patients older than 65 years of age is related to the presence of a greater number of comorbidities.

According to our and other studies, the incidence of CVC development in RA patients under TCZ therapy did not exceed that in patients not receiving TCZ. In RCTs investigating the safety of TCZ use [36], the incidence of MI was 0.25 cases per 100 patient-years, and stroke was 0.19 cases per 100 patient-years. In the ICHIBAN study [76], the incidence of MI and acute coronary syndrome was 0.7 cases per 100 patient-years, and stroke was 0.4 cases per 100 patient-years. G. Jones et al. [77] observed 1.21 cases of CVCs per 100 patient-years over 276 weeks.

An analysis of the US MarketScan database, which included more than 15,000 patients with RA who had not previously received TCZ, revealed a similar incidence of CVCs: MI: 0.8 events per 100 patient-years; stroke: 0.51 events per 100 patient-years [78].

The involvement of IL-6 in the pathogenesis of CVDs has led to clinical trials investigating drugs targeting IL-6 and its receptor, which are well established in the treatment of RA. The phase II RESCUE clinical trial demonstrated the effect of ziltivekimab, an IL-6 antibody blocker, on reducing CRP in patients with high risk of atherosclerotic CVD [5,79]. The ASSAIL-MI-trial showed that TCZ limited the area of myocardial necrosis in patients with acute ST-segment elevation MI [80]. The ongoing randomized controlled phase III ZEUS trial aims to establish the feasibility of ziltivekimab to reduce the incidence of CVCs in patients with atherosclerotic cardiovascular diseases and chronic kidney disease [81]. The eagerly awaited ARTEMIS trial (NCT06118281), which starts this year, will study the effect of ziltivekimab in patients after acute myocardial infarction. The results of the ongoing randomized controlled ZEUS trial, which aims to establish whether ziltivekimab can be used to reduce the incidence of SSO in patients with ACCH and CKD [81], are also eagerly awaited. Moreover, starting this year is the new ARTEMIS trial (NCT06118281), which aims to study the effects of ziltivekimab in patients after acute MI.

## 4. Materials and Methods

Thirty-seven patients with RA determined according to the 2010 American College of Rheumatology/European Alliance of Rheumatologic Associations (ACR/EULAR) criteria were included. All patients were observed at the V.A. Nasonova Research Institute of Rheumatology, Moscow, Russia from 2015 to 2022. This study was performed in accordance with the Declaration of Helsinki of 1975 and its revised version (2013). The study protocol was approved by the Local Ethics Committee of the Nasonova Research Institute of Rheumatology № 11 on 2 February 2015 and № 13 on 10 February 2022. All patients signed informed consent for participation in this study.

This study did not include people over 70 years old; people with CHF III-IV functional class NYHA, severe chronic diseases (cancer, renal and hepatic failure), and conditions that prevent biological therapy were also excluded.

All patients were evaluated for traditional cardiovascular RFs. The modified Systematic Coronary Risk Evaluation (mSCORE) with the adjustment (×1.5) recommended by EULAR for RA patients was used to calculate the CVR [82].

The ultrasound examination of CA was carried out on the ultrasound system Esaote MyLab Twice (Italy). Atherosclerotic lesions of CA were assessed by the detection of atherosclerotic plaques (ASPs) determined as a local increase in CA by more than 50% compared to surrounding areas or when the thickness of the intima–media complex of CA (cIMT) was greater than 1.5 mm with protrusion towards the vessel lumen.

The concentrations of total cholesterol, HDL-C, and TG were determined by standard enzymatic methods. The level of LDL-C was calculated using the Friedwald formula: LDL-C = total cholesterol − TG/5 − HDL-C. The atherogenicity index (AI) was calculated according to the following formula: AI = (total cholesterol − HDL)/HDL. The levels of CRP and IgM-RF in blood serum were measured by the immunonephelometric method using a BN Pro Spec analyzer Siemens Ltd., Germany. The anti-CCP antibody concentration was determined by enzyme immunoassay using a commercial kit from Axis-Shield Diagnostics Ltd. (Dundee, UK). The concentrations of sVCAM-1 and sICAM-1 in venous blood serum were determined using the enzyme-linked immunosorbent assay (ELISA), using reagent kits and according to the protocols of Bender MedSystems (Burlingame, CA, USA). The IL-6 concentration was measured using a commercial Human IL-6 DuoSet ELISA kit (R&D Systems Inc., Minneapolis, MN, USA).

Statistical processing of data was performed using the program Statistica 12. Absolute and relative values were presented for qualitative signs; median, 25th, and 75th percentiles were presented for quantitative signs. The Mann–Whitney test was used to compare two independent groups for quantitative characteristics, and the χ^2^ test was used for qualitative characteristics (with Yates correction for a small number of observations). The correlation between signs was evaluated using the Spearman rank correlation criterion (R). Differences were considered statistically significant at *p* < 0.05.

## 5. Conclusions

During the 5-year follow-up period of RA patients on TCZ therapy, no increases in CVR, cIMT, the incidence of traditional RFs, carotid ASPs, and CVCs were revealed. Statin therapy successfully controlled dyslipidemia in patients on long-term TCZ treatment. In the group receiving statins, a significant decrease in AI was found and blood lipoprotein content correlated with parameters of disease activity.

## Figures and Tables

**Figure 1 ijms-25-12425-f001:**
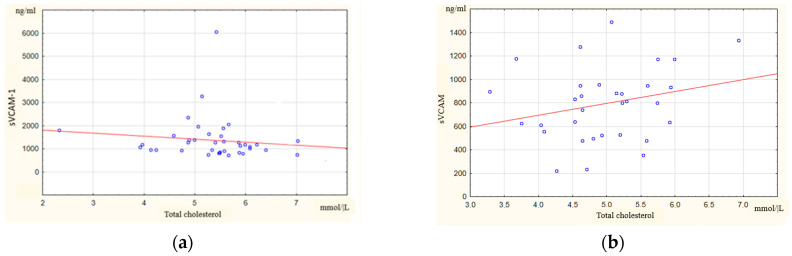
Correlation of total cholesterol and sVCAM-1 in RA patients with carotid artery atherosclerosis at baseline (**a**) and at the end (**b**) of this study.

**Table 1 ijms-25-12425-t001:** General characteristics of patients with RA included in this study (*n* = 45).

Parameter	Value
Age, years, Me (25;75 percentile)	54 (46;66)
Gender, women/men, *n* (%)	38 (84)/7 (16)
Duration of disease, months, Me (25;75 percentile)	90 (47;156)
Stage, *n* (%):	
Early	7 (16)
Advanced	14 (31)
Late	24 (63)
Extra-articular manifestations, *n* (%)	27 (60)
DAS28-ESR, Me (25; 75 percentile)	6.2 (5.5;6.8)
SDAI, Me (25; 75 percentile)	35 (29;41)
CDAI, Me (25; 75 percentile)	31 (24;37)
HAQ, Me (25; 75 percentile)	1.75 (1.25;2.125)
IgM RF +, *n* (%)	45 (100)
Anti-CCP antibodies +, *n* (%)	38 (84)
DMARD intolerance, *n* (%)	11 (45)
Inefficacy of two or more DMARDs, *n* (%)	37 (82)
Methotrexate + leflunomide + sulfasalazine, *n* (%)	7 (16)
Methotrexate + leflunomide, *n* (%)	15 (33)
Methotrexate + sulfasalazine, *n* (%)	15 (33)
Glucocorticoids, *n* (%)	23 (51)
NSAIDs, *n* (%)	30 (67)
Statins, *n* (%)	7 (15)

Note: DAS28—disease activity score; ESR—erythrocyte sedimentation rate; SDAI—simplified disease activity index; CDAI—clinical disease activity index; HAQ—health assessment questionnaire; RF—rheumatoid factor; anti-CCP antibodies—anti-citrullinated protein antibodies; DMARDs—disease-modifying antirheumatic drugs; and NSAIDs—nonsteroidal anti-inflammatory drugs.

**Table 2 ijms-25-12425-t002:** Dynamics of the main indicators of inflammatory and immunologic activity, Me (25;75 percentiles).

Parameter	Patients (*n* = 37)	
	Before Treatment	After 260 Weeks
DAS28-ESR	6.2 (5.5;6.8)	2.0 (1.3;2.9) *
SDAI	35 (29;41)	2.8 (1.9;3.5)
CDAI	31 (24;37)	2.1 (1.5;3.1)
HAQ	1.75 (1.25;2.125)	0.5 (0.25;1.0) *
IgM-RF, IU/mL	221.0 (40.2;568.4)	77.9 (23.9;282.8)
Anti-CCP antibodies, units/mL	300 (30;300)	285 (32;300)
CRP, mg/L	29.0 (11.0;80.8)	0.3 (0.1;2.1) *
ESR, mm/h	48 (30;68)	5 (3;10) *
sVCAM-1, ng/mL	1721 (997;1921)	851 (611;1535) *
sICAM-1, ng/mL	321 (270;411)	298 (217;402)
Serum IL-6, pg/mL	79 (68;92)	7 (5;8) *

Note: *—*p* < 0.01. RF—rheumatoid factor; anti-CCP antibodies—anti-citrullinated protein antibodies; CRP—C-reactive protein; ESR—erythrocyte sedimentation rate; sVCAM-1—soluble vascular cell adhesion molecule-1; sICAM-1—soluble intercellular adhesion molecule-1; and IL—interleukin.

**Table 3 ijms-25-12425-t003:** Dynamics of the frequency of CVD RFs and BMI in RA patients over 265 weeks of TCZ therapy.

Parameter	Patients (*n* = 45)	
	Before Treatment	After 260 Weeks
Dyslipidemia, *n* (%)	30 (67)	23 (51)
Total cholesterol > 5.0 mmol/L, *n* (%)	30 (67)	8 (18) *
HDL-C <1.0 mmol/L in men or <1.2 mmol/L in women, *n* (%)	15 (33)	3 (7) *
LDL-C > 2.6 mmol/L, *n* (%)	30 (67)	23 (51)
TG >1.7 mmol/L, *n* (%)	9 (20)	14 (31)
Arterial hypertension, *n* (%)	29 (64)	30 (67)
Body weight deficiency, *n* (%)	6 (13)	-
Overweight, *n* (%)	23 (51)	28 (62)
BMI, kg/m^2^, Me (25; 75 percentiles)	26.1 (21.8;33.2)	29.1 (27.5;35.1) *
Family history of CVD, *n* (%)	16 (36)	-
Smoking, *n* (%)	7 (16)	3 (7)

Note: *—*p* < 0.05. HDL-C—high-density lipoprotein cholesterol; LDL-C—low-density lipoprotein cholesterol; TG—triglyceride; BMI—body mass index; and CVD—cardiovascular disease.

**Table 4 ijms-25-12425-t004:** Dynamics of blood lipid spectrum parameters in RA patients of groups 1 and 2, Me [25;75 percentile].

Parameter	Group 1 (*n* = 21)		Group 2 (*n* = 24)	
	Before Treatment	After 265 Weeks	Before Treatment	After 265 Weeks
Total cholesterol, mmol/L	5.3 (4.5;6.1) *	4.5 (3.1;4.1) *	4.9 (4.4;6.2)	5.2 (4.6;6.8)
HDL-C, mmol/L	1.5 (1.1;1.7) *	2.1 (1.6;2.3) *	1.6 (1.3;1.8)	1.8 (1.3;2.0)
LDL-C, mmol/L	3.9 (2.8;3.4)	2.9 (2.4;3.9)	3.4 (2.8;3.6)	3.0 (2.1;4.2)
TG, mmol/L	1.3 (0.9;1.7)	1.1 (0.8;1.5)	1.1 (0.9;1.4)	1.5 (1.1;1.9)
Atherogenicity index	2.7 (2.1;3.2) *	1.2 (1.0;2.7) *	2.2 (1.8;2.6)	1.7 (1.3;2.9)

Note: *—*p* < 0.01.

**Table 5 ijms-25-12425-t005:** Frequency of carotid ASPs and dynamics of cIMT in RA patients (*n* = 45).

Parameter	Before Treatment	After 265 Weeks
Carotid ASPs, %:		
None	33	28
1 carotid ASP	54	54
≥2 carotid ASPs	13	18
cIMT max left, mm	0.9 (0.7;0.9)	1.1 (0.8;1.4)
cIMT max right, mm	0.8 (0.6;0.9)	1.0 (0.8;1.1)

Note: cIMT—carotid intima–media thickness; ASP—atherosclerotic plaque.

## Data Availability

Data are available from the corresponding author upon request.

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
