# Peer review of "Interleukin-6: Cardiovascular Aspects of Long-Term Cytokine Suppression in Patients with Rheumatoid Arthritis"

_ijms, 2024, doi:10.3390/ijms252212425_

Round 1
Reviewer 1 Report
Comments and Suggestions for Authors
This study evaluates the long-term effects of TCZ on cardiovascular risk and structural changes in the carotid arteries in RA patients. The findings suggest that while TCZ effectively reduces disease activity, the cardiovascular risk factors remain stable, with some improvements in lipid profiles and a decrease in atherosclerotic plaque formation over a 260-week period. Although the sample size was relatively small, the long-term focus on both inflammatory control and cardiovascular health in RA patients provides valuable insights, especially regarding the interplay between IL-6 inhibition and cardiovascular outcomes. The study uses a variety of biomarkers and clinical parameters, including lipid profiles, carotid artery assessments, and inflammatory markers, ensuring robust and detailed data. However, some issues should be addressed.
1. Introduction
I think it should be added that a well-known side effect of TCZ is the increase in LDL, TG, etc.
2. Results
why did the authors measure vCAM and ICAM? if needed, the authors should describe the importance of measuring these in the introduction.
3. Results
DAS28 was based on ESR or CRP? DAS28-ESR or DAS28-CRP?
4. Materials and methods.
Table 5 should be included in the Results section.
5. Discussion
The manuscript could further enhance its discussion by comparing the findings with previous studies on IL-6 inhibition and cardiovascular outcomes in RA.
Author Response
Dear reviewer,
We thank you for the work you have done and your interest in the study.
All your comments have been taken into account and corrected. Thanks to them, the manuscript has been improved.
- Introduction
I think it should be added that a well-known side effect of TCZ is the increase in LDL, TG, etc.
Answer: Thank you for your comment, we have corrected it.
- Results
why did the authors measure vCAM and ICAM? if needed, the authors should describe the importance of measuring these in the introduction.
Answer: Your comment has been taken into account, we have corrected this remark.
- Results
DAS28 was based on ESR or CRP? DAS28-ESR or DAS28-CRP?
Answer: Thank you for your comment. RA activity in the study was determined by DAS28-ESR. A correction has been made to the text.
- Materials and methods.
Table 5 should be included in the Results section.
Answer: Table 5 has been included in the Results section.
- Discussion
The manuscript could further enhance its discussion by comparing the findings with previous studies on IL-6 inhibition and cardiovascular outcomes in RA.
Answer: We added studies on IL-6 inhibition and cardiovascular outcomes in RA and compared the results.
Reviewer 2 Report
Comments and Suggestions for Authors
The manuscript by Elena V. Gerasimova shows extensive follow up analysis of lipid profile and cardiovascular disease of Rheumatoid Arthritis (RA) patients treated with a humanized monoclonal antibody blocking IL-6 receptor. The data shown looks solid although the number of patients enrolled in this study is not very high. It is already known that inflammation can promote the formation of arteriosclerotic plaques but not all the anti-inflammatory drugs or monoclonal antibody therapies have positive effect on the cardiovascular network, at variance some contribute to cardiovascular disease. In the case of tocilizumab (TCZ) authors show that therapy does not alter cardiovascular profile nor the lipid profile but substantially ameliorates RA disease score and various inflammatory serologic markers.
I sustain the publication on its present form.
Author Response
Dear Reviewer,
We thank you for your work, your interest in the study and your high evaluation of our manuscript.
This means that we are on the right track and are ready to continue our observational study.
Round 2
Reviewer 1 Report
Comments and Suggestions for Authors
The authors revised the manuscript based on my comments.